Gastrocnemius fascicles are shorter and more pennate throughout the first month following acute Achilles tendon rupture

http://orcid.org/0000-0003-3625-1082 Hullfish Todd J.
O’Connor Kathryn M.
http://orcid.org/0000-0003-0269-2170 Baxter Josh R. josh.baxter@uphs.upenn.edu
Department of Orthopaedic Surgery, University of Pennsylvania , Philadelphia, PA , USA
Hutchinson John
Electronic publication date: 2019 Apr 23
Publication date: 2019
Volume: 7
Electronic Location ID: e6788
Received 2018 Nov 29; Accepted 2019 Mar 14
Copyright: © 2019 Hullfish et al.
Copyright year: 2019
Copyright holder: Hullfish et al.
License: This is an open access article distributed under the terms of the Creative Commons Attribution License, which permits unrestricted use, distribution, reproduction and adaptation in any medium and for any purpose provided that it is properly attributed. For attribution, the original author(s), title, publication source (PeerJ) and either DOI or URL of the article must be cited.
License URL: https://creativecommons.org/licenses/by/4.0/

Keywords: Muscle, Remodeling, Pennation, Gastrocnemius, Plantarflexors, Achilles tendon, Rupture, Fascicle, Ultrasound

Funding: Thomas B. McCabe and Jeannette E. Laws McCabe Fund This work was supported by the Thomas B. McCabe and Jeannette E. Laws McCabe Fund. The funders had no role in study design, data collection and analysis, decision to publish, or preparation of the manuscript.

==============================
The purpose of this study was to characterize the short-term effects of Achilles tendon ruptures on medial gastrocnemius. We hypothesized that the fascicles of the medial gastrocnemius muscle of the injured Achilles tendon would be shorter and more pennate immediately following the injury and would persist throughout 4 weeks post-injury. B-mode longitudinal ultrasound images of the medial gastrocnemius were acquired in 10 adults who suffered acute Achilles tendon ruptures and were treated non-operatively. Ultrasound images were acquired during the initial clinical visit following injury as well as 2 and 4 weeks following this initial clinical visit. Resting muscle structure was characterized by measuring fascicle length, pennation angle, muscle thickness, and muscle echo intensity in both the injured and contralateral (control) limbs. Fascicle length was 15% shorter (P < 0.001) and pennation angle was 21% greater (P < 0.001) in the injured muscle compared to the uninjured (control) muscle at the presentation of injury (week 0). These differences in fascicle length persisted through 4 weeks after injury (P < 0.002) and pennation angle returned to pre-injury levels. Muscle thickness changes were not detected at any of the post-injury visits (difference < 4%, P > 0.026). Echo intensity of the injured limb was 8% lower at the presentation of the injury but was not different compared to the contralateral muscle at 2 and 4 weeks post-injury. Our results suggest that Achilles tendon ruptures elicit rapid changes in the configuration of the medial gastrocnemius, which may explain long-term functional deficits.

Introduction

The incidence of Achilles tendon ruptures has increased 10-fold over the last three decades, disproportionally affecting athletic adults who participate in sports requiring sudden acceleration and jumping (Lantto et al., 2015). This well-documented increase in the prevalence of acute Achilles tendon ruptures (Nyyssönen, Lüthje & Kröger, 2008; Huttunen et al., 2014; Lantto et al., 2015) is most likely explained by the recent rise in sports participation amongst aging adults (Huttunen et al., 2014). While advances in rehabilitation protocols have improved outcomes in patients treated either operatively or non-operatively (Willits et al., 2010), approximately one out of five patients without other complications are unable to return to their previous levels of athletic participation (Zellers, Carmont & Silbernagel, 2016) and two out of three have long-term functional deficits (Brorsson et al., 2017).

Deficits in plantarflexion function of the affected limb play a large role in preventing return to pre-injury levels of activity (Olsson et al., 2011) and have been shown to persist as far out as 14-years following injury (Heikkinen et al., 2016; Brorsson et al., 2018). Clinically, these deficits present as resting ankle angles that are less plantarflexed (Zellers, Carmont & Silbernagel, 2018), reduced heel-raise height (Silbernagel, Steele & Manal, 2012; Brorsson et al., 2017), and 10–20% reductions in plantarflexion strength (Leppilahti et al., 2000; Pajala et al., 2009; Heikkinen et al., 2016). Tendon elongation and plantarflexor muscle atrophy have both been associated with these reductions in function (Mullaney et al., 2006; Silbernagel, Steele & Manal, 2012; Heikkinen et al., 2017a). Decreased resting tension in the muscle-tendon unit stimulates sarcomere subtraction in small animals (Williams & Goldspink, 1978; Williams, 1990), effectively restoring the resting tension in the muscle. Both increased tendon length and decreased muscle fascicle length have been shown to negatively affect single-leg heel raise performance in computational simulations (Baxter, Farber & Hast, 2018).

Plantarflexor function is governed by muscle–tendon structure. Although isometric strength is a commonly measured metric of patient function and correlates with muscle size and leverage (Baxter & Piazza, 2014), athletics require active plantarflexor shortening to generate ankle power (Lee & Piazza, 2009). Longer muscle fascicles generate greater power during isokinetic contractions (Drazan, Hullfish & Baxter, 2019b), which is beneficial for activities like sprinting (Lee & Piazza, 2009). Targeted training can increase muscle fascicle length (Salzano et al., 2018), effectively increasing power potential. Although muscle structure has been linked to function in many populations, shorter muscle fascicles have only been reported in a single case report of a patient with poor outcomes following an Achilles tendon rupture (Baxter, Hullfish & Chao, 2018).

Sudden changes in muscle-tendon tension, either induced through joint immobilization (Williams & Goldspink, 1978; Williams, 1990) or changes in muscle leverage (Burkholder & Lieber, 1998; Koh & Herzog, 1998), stimulate changes in muscle fascicle length within weeks. While a previous report documented muscle remodeling 3–6 months following acute Achilles tendon ruptures (Peng et al., 2017), the response to an immediate loss of muscle-tendon tension caused by Achilles tendon ruptures has not been prospectively studied in a patient cohort. Other studies have found that 3 weeks of immobilization stimulates decreases in muscle strength by as much as one-half in humans (Hortobágyi et al., 2000) and muscle remodeling characterized by a 25% decrease in serial sarcomere count (Williams & Goldspink, 1978).

Therefore, the purpose of this study was to quantify the immediate structural changes to the medial gastrocnemius in patients who suffered acute Achilles tendon ruptures and were treated non-operatively. We decided to study the medial gastrocnemius muscle because our previous work found large changes in its structure that explained long-term functional deficits (Baxter, Hullfish & Chao, 2018). We hypothesized that the medial gastrocnemius muscle would have shorter and more pennate fascicles immediately following an acute Achilles tendon rupture and these changes would persist throughout 4 weeks post-injury. Based on previous reports (Williams & Goldspink, 1978; Hortobágyi et al., 2000), we expected that changes in muscle structure following acute Achilles tendon rupture should be detected within the first 4 weeks following injury.

Methods

Study design

Ten adults (nine males, Age: 44 ± 12; BMI: 28.6 ± 6.5) who suffered acute Achilles tendon ruptures (seven suffered during sports) and were treated non-operatively by a fellowship trained foot and ankle surgeon and provided written-informed consent in this study approved by the University of Pennsylvania IRB (828374). All patients were recruited from the clinics of the Department of Orthopaedic Surgery at the Penn Medicine and met several inclusion criteria: patient was between 18 and 65 years old, elected to be treated non-operatively for acute Achilles tendon rupture within 2 weeks of injury. Patients were not enrolled in this study if they met any of our exclusion criteria: excessive weight (BMI < 50) or concomitant lower extremity injuries. These patients were enrolled in a 1-year prospective study aimed at linking muscle remodeling with patient outcomes and function. To first determine how muscle structure changes in the first month following injury, we analyzed the imaging data of all patients enrolled in this study at the time of analysis. We acquired ultrasound images of the medial gastrocnemius during three clinical visits (Fig. 1): time of injury when patients were placed in a cast (week 0); time of cast removal when patients were transitioned into partial weight bearing in a walking boot (week 2); and 2 weeks after transitioning into boot use (week 4). Patients were enrolled in this study within 10 days of suffering the injury (4.5 ± 3.5 days), at which the week 0 images were acquired. We imaged both the affected and contralateral limbs at each time point to determine changes in resting muscle structure. Resting structure of the muscle was characterized by the length and pennation angle of the constituent fascicles as well as the thickness of the muscle belly. We also performed a secondary analysis of average muscle echo intensity to characterize “muscle quality”, which may be a surrogate measure of muscle remodeling and atrophy (Fukumoto et al., 2012).

Figure 1 Study outline.

Patients who suffered acute Achilles tendon ruptures and received non-operative treatment were enrolled in this prospective study at their initial clinical visit (week 0). We measured the structure of the medial gastrocnemius muscle on both the injured and contralateral sides using ultrasound at each clinical visit (weeks 0, 2, and 4). At initial presentation, patients were placed in a plantarflexed cast and were instructed to avoid any weight bearing. At the 2-week follow-up visit, the cast was removed and patients were placed in an orthopedic boot with heel wedges to place the foot in approximately 30 degrees of plantarflexion. Patients were instructed to apply light pressure to the foot while ambulating with crutches.

Image acquisition

Longitudinal images of the medial gastrocnemius (Fig. 2) were acquired while patients lay prone on a treatment table with their feet and ankles supported by the edge of the table in plantarflexion and kept in the same position for all imaging sessions. At the beginning of each imaging session, patients lay on the treatment table with their ankles resting off the edge of the table (Zellers, Carmont & Silbernagel, 2018). We have used this resting angle previously to longitudinally study Achilles tendon structure (Hagan et al., 2018). In order to minimize the loads applied to the healing tendon, the patients slid proximally on the table to support both ankles in the resting position of the uninjured limb. Continuous B-mode ultrasound images of the medial gastrocnemius were acquired by a single investigator using an eight MHz ultrasound transducer with a six cm scanning width (LV7.5/60/128Z-2, SmartUs, TELEMED). We followed the guidelines for reliably imaging medial gastrocnemius muscles outlined by Bolsterlee, Gandevia & Herbert (2016). Briefly, we positioned the probe half way between the muscle-tendon junction and crease of the knee in the central portion of the muscle belly and aligned with the fascicles to ensure that the muscle fascicles lie in the image plane. During pilot testing, we identified scanning parameters (scan parameters: Dynamic Range: 72 dB; Frequency: 8 MHz; Gain: 47 dB) that produced high-contrast images and held them constant for all patients and scanning sessions. We also confirmed during pilot testing that imaging along the midline of the muscle belly produced similar measurements of fascicle length and pennation angle. Specifically, imaging the central, medial, lateral, medial-central, and latera-central regions of the muscle resulted in similar measurements of fascicle length (standard deviation < 1.5 mm) and pennation angle (standard deviation < 1 degree).

Figure 2 Ultrasound imaging quantifying muscle structure.

Ultrasound images of the medial gastrocnemius were acquired and analyzed by the same investigator to quantify resting fascicle length, pennation angle, thickness, and muscle belly echogenicity. This figure shows a representative image of both the uninjured (A, control) and injured (B) sides. Fascicle length and pennation angle was analyzed by the same observer for all trials. Muscle thickness was calculated as the product of the fascicle length and sine of the pennation angle. During our initial analyses, we noticed that the injured limb displayed “poor quality” muscle, as noted by a lack of contrast between the fascicles and interstitial connective tissue.

All images were acquired by the same investigator and saved as digital videos to be processed later by the same investigator. During each imaging session, the investigator pressed a foot switch when the ultrasound image produced fascicles that were in plane in order to analyze these specific frames during post-processing. These individual frames from the continuous imaging were exported as still images to be processed for muscle structure. In order to protect the rupture, we acquired images with the foot fully supported in plantarflexion. During the first study visit (week 0), the Achilles tendon did not transfer any load, which was confirmed by moving the ankle joint while observing no change in muscle structure via real-time ultrasound imaging. Because of this observation, we considered the first visit scans of the injured muscle, which were acquired on average 4.5 days after injury, to accurately represent muscle structure immediately following the rupture. To confirm that probe placement was repeatable between sessions, we imaged the contralateral (control) muscle at each visit and found that these measurements of muscle architecture had coefficients of variation of less than 10% and intraclass correlations greater than 0.84 (Supplemental Material). These metrics of reliability agree with previous reports in the literature (Kwah et al., 2013).

Image analysis

Resting fascicle length, pennation angle, muscle thickness, and muscle echo intensity were quantified using custom-written software (MATLAB 2017b; The MathWorks, Inc., Natick, MA, USA) (Baxter, Hullfish & Chao, 2018). We anonymized and randomized each image to ensure that the investigator analyzing the images could not be biased. For each image, the investigator identified the deep and superficial aponeuroses as well as a single fascicle (Fig. 2). We quantified fascicle length as the distance between the fascicle’s insertions into the aponeuroses. Pennation angle was determined to be the angle between the fascicle and the deep aponeuroses. Muscle thickness was calculated based on fascicle length and pennation angle (Eq. 1). During this analysis, we observed that many of the ultrasound images showed the muscle to be of “poor quality”, which was visually apparent based on reduced contrast between muscle fascicle and interstitial connective tissue (Fig. 2B). Therefore, we calculated the mean echo intensity between deep and superficial aponeuroses to quantify muscle quality. Because we acquired all ultrasound images with the same scanning parameters, we used the raw echo intensity values that ranged between 0 and 255. A previous study correlated muscle strength with mean echo intensity and attributed this link between muscle form and function to increased fibrous and adipose tissues within the muscle belly (Fukumoto et al., 2012).

(1) Thickness=lFascicle × sinθPennation

Statistical analysis

To test our hypothesis that resting muscle architecture would change following injury, resting fascicle length, pennation angle, muscle thickness, and mean echo intensity at weeks 0, 2, and 4 were each compared to against measurements of the contralateral muscle at each visit using paired one-way t-tests. We adjusted for multiple comparisons using a Bonferroni correction, which set the threshold for statistical significance to P = 0.05/3 = 0.0167, where 3 is the number of comparisons for each variable of interest. When muscle structure parameters differed from the control values, we calculated the Cohen’s effect size (d). We performed an a priori sample size calculation based on the variation of medial gastrocnemius fascicle length in young adults (Baxter & Piazza, 2014) and determined that 10 patients would be able to detect a 10% decrease with desired statistical power of 0.8.

Results

Gastrocnemius muscle structure following an acute Achilles tendon rupture differed with the healthy-contralateral muscle throughout the first 4 weeks following injury (Fig. 3). Fascicle length was 15% shorter (d = 1.7, P < 0.001) and pennation angle was 21% greater (d = 1.6, P < 0.001) at the presentation of injury (week 0). These differences in fascicle length persisted throughout the 4 weeks after the injury (d > 1.6, P < 0.001) while pennation angle was similar at week 2 (d = 0.9, P = 0.017) before returning to within 10% of the contralateral muscle at week 4. Muscle thickness changes were not detected at any of the post-injury visits; however, a 3–4% decrease in muscle thickness at weeks 2 and 4 approached our threshold for significance when controlling for multiple comparisons (d = 0.7–0.8, P = 0.026–0.28). Muscle quality, measured as mean echo intensity of the muscle belly, was 8% lower in the injured limb immediately (d = 0.9, P = 0.008) and 11% lower 2 weeks following injury (d = 1.5, P < 0.001). At week 4 muscle quality had returned to within 1% of the contralateral limb (P = 0.393).

Figure 3 Muscle structure following Achilles tendon rupture.

Medial gastrocnemius structure following Achilles tendon rupture (closed circles) was compared to the contralateral muscle (control - crosses) at each time point. Resting fascicle length (A) decreased while pennation angle (B) increased in similar proportions, which explained the stable muscle belly thickness (C) following 4 weeks of injury. Muscle echogenicity (D) decreased by 8% at week 0 but returned to non-injured values at weeks 2 and 4. (*P < 0.017, dashed lines—95% confidence intervals, d—Cohen’s effect size, %Δ—percent change from control data, week 0 – N = 10, week 2 – N = 8, week 4 – N = 10).

Discussion

The purpose of this study was to quantify medial gastrocnemius structure following the first month of acute Achilles tendon ruptures. Our findings support our hypothesis that the gastrocnemius muscle fascicles of the affected side would demonstrate shorter length and greater pennation angle than the contralateral control muscle. These changes appear to be a mechanical response to a sudden loss of intra-muscular tension—typically present with an intact tendon (Hug et al., 2013)—that likely stimulates biological remodeling of the serial sarcomeres observed in immobilization studies (Williams & Goldspink, 1978). We confirmed that no load was carried between the Achilles tendon and gastrocnemius following injury by manually rotating the ankle while observing no change in muscle fascicle length or pennation angle via real-time ultrasound imaging. Our report is the first to our knowledge to document gastrocnemius remodeling within the first month following an Achilles tendon rupture.

Our measurements of medial gastrocnemius structure compared favorably with prior reports. Due to the clinical constraints of these patients, we were required to image the muscle of the injured side with the foot supported in maximal plantarflexion by a table. As expected, our measurements of gastrocnemius structure were shorter and more pennate than compared to other studies that have measured gastrocnemius structure at neutral ankle angle (Lee & Piazza, 2009; Baxter & Piazza, 2014). Unlike these previous studies, we measured muscle architecture with the ankle fully supported in plantarflexion to protect the healing tendon. As expected, our measurements of fascicle length and pennation angle were similar to a previous report of gastrocnemius structure in peak plantarflexion (Gao et al., 2009), which demonstrated a linear decrease in fascicle length and increase in pennation angle with increasing plantarflexion angle. We calculated muscle thickness using a single longitudinal scan of the muscle belly rather than a transverse scan or multiple longitudinal scans at different locations. Using this imaging approach, we detected a small (4%) decrease in muscle thickness at week 4 that approached statistical significance (P = 0.026). This small decrease in muscle thickness was less than previous reports of muscle atrophy at least 1 year following injury (Heikkinen et al., 2017a, 2017b). Our findings that resting fascicle length decreases following Achilles tendon rupture also compare favorably with a previous report that found fascicle length decreases by 20% 3–6 months after Achilles tendon ruptures were surgically repaired (Peng et al., 2017). Decreases in muscle quality have been associated with age-related muscle deterioration such as sarcopenia (Fukumoto et al., 2012), but these changes have not been observed in muscle following tendon rupture. Side to side differences in rectus femoris fascicle length, pennation angle, thickness, and echo intensity have been reported to be less than 11% (Mangine et al., 2014). However, this previous study quantified the muscle of elite basketball players that may have had greater muscle structure asymmetry due to sport specific training and history.

Functional deficits are associated with elongated tendon and shorter muscle fascicles following Achilles tendon rupture. Recent reports and our current findings document changes in plantarflexor structure that occurs within the first weeks following rupture, persist through 3–6 months (Peng et al., 2017), and appear to lead to permanent muscle remodeling (Baxter, Hullfish & Chao, 2018). Tendon elongation has been identified as a key clinical measure that explains long-term deficits in function (Silbernagel, Steele & Manal, 2012; Brorsson et al., 2017). When considered in this context, our current findings of shorter and more pennate gastrocnemius fascicles immediately following Achilles tendon rupture indicate a coupled remodeling response with the healing tendon (Suydam et al., 2015). Imaging studies of the healing Achilles tendon show continued tendon elongation (Nyström & Holmlund, 1983; Mortensen, Skov & Jensen, 1999; Kangas et al., 2007), which suggests that muscle-tendon adaptations continue for several months following the injury. We hypothesize that the initial tendon injury mechanically unloads the muscle that stimulates a rapid change in muscle structure in an attempt to restore the optimal resting tension in the muscle-tendon unit (Williams & Goldspink, 1978), which potentially triggers a positive feedback loop. Prospective research is needed to directly link the interaction between tendon elongation and muscle remodeling. Understanding the progression of this remodeling process is crucial for the treatment of tendon injuries and can be used to inform clinical decisions as well as the prescription of rehabilitation protocols. Although tendon loading is restricted during the first month following injury, shorter and more pennate muscle fascicles may lead to increased muscle-tendon tension and drive additional tendon elongation (Kangas et al., 2007).

In addition to our current findings that skeletal muscle architecture is sensitive to tendon rupture, muscle fascicle length can also be affected by loading, pathology, surgical procedures, and immobilization in both humans and animal models. High-acceleration training during maturation in guinea fowl stimulates longer muscle fascicles that contain greater amounts of sarcomeres in series (Salzano et al., 2018). Children with cerebral palsy have shorter gastrocnemius fascicles that can be increased with surgical correction of the plantarflexor contracture (Wren et al., 2010). Increasing the muscle shortening demands by surgically releasing the ankle retinaculum elicits rapid sarcomere subtraction in both mice (Burkholder & Lieber, 1998) and rabbits (Koh & Herzog, 1998). Immobilizing the ankle joint in plantarflexion elicits muscle fiber remodeling, highlighted by a shorter optimal length and reduced sarcomeres in series (Williams & Goldspink, 1978; Williams, 1990). Seminal work by Williams (1990) demonstrates the importance of muscle tension on muscle length and joint range of motion in the context of immobilization. Achilles tendon ruptures have been associated with muscle atrophy (Rosso et al., 2013; Heikkinen et al., 2017a); however, we did not quantify any decreases in muscle thickness throughout the 4 weeks following the injury. While our measurements of pennation angle made during the first 2 weeks following injury demonstrate large increases in pennation angle (effect size > 0.9), the 95% confidence intervals at 4 weeks suggests that the muscle pennation angle decreased. These observations agree with previous reports that found no difference in pennation angle at 6 months between the injured and uninjured sides (Peng et al., 2017). This improvement in pennation angle may be due to increased weight bearing when patients transition from the full-leg cast to the walking boot. However, additional research is required to fully understand the tendon loading biomechanics during ambulation in these two protective devices. While our study was not designed to quantify muscle volume, it is plausible that atrophy did occur during our time points when considering a shorter muscle belly defined by shorter and more pennate fascicles.

This study had several limitations, which should be considered when interpreting our findings. We focused on the first 4 weeks following acute Achilles tendon rupture to characterize structural changes to the skeletal muscle. While this timespan may appear short, previous immobilization studies in both humans (Hortobágyi et al., 2000) and small animals (Williams & Goldspink, 1978) have demonstrated that this time scale elicits detectable changes to both muscle structure and function. We imaged the gastrocnemius muscle with both ankles supported in their resting position and did not control this angle between test sessions. However, our repeated imaging of the control limb suggests that this ankle angle was consistent because measures of muscle structure of the control limb were similar between sessions. This study only tested the effects of non-operatively treated Achilles tendon ruptures, which limits the implications of our findings to that specific population of patients. Out of the 10 patients enrolled in this study, only one was female, which is consistent with the four to one ratio of male to female patients reported in the literature (Raikin, Garras & Krapchev, 2013). We did not perform muscle biopsy or intramuscular sarcomere measurement (Sanchez et al., 2015); as such, this study was not designed to establish whether optimal sarcomere length was preserved via sarcomere subtraction or not. However, animal studies that imposed sudden changes in joint mechanics demonstrate that rapid change in sarcomere number to preserve optimal sarcomere length (Williams, 1990; Burkholder & Lieber, 1998; Koh & Herzog, 1998). Additionally, our measurements of “muscle quality” that were quantified by average echo intensity have been described in the literature (Fukumoto et al., 2012) but should be investigated further to validate this measurement. This qualitative decrease “muscle quality” may have affected our manual measurements of fascicle length and pennation angle. However, our group has demonstrated that manual measurements of healthy muscle under passive and maximal contractions are highly repeatable (Drazan, Hullfish & Baxter, 2019a), the latter having a similar appearance to patients with injured Achilles tendon. We decided to image the medial gastrocnemius, which has longer and less pennate fascicles than the soleus and generates tension in greater amounts of plantarflexion (Hirata et al., 2015), which is critical for functional tasks such as heel raises. We did not measure muscle structure prior to the injury, and therefore it possible that these structural differences existed prior to the injury. However, muscle atrophy is well reported in this population (Heikkinen et al., 2017a), suggesting that muscle structure was symmetrical prior to injury. Controlled experiments using small animal models are needed to test the effects of pre-injury muscle structure on post-injury remodeling.

These findings are a preliminary data set from a larger clinical cohort of patients that were enrolled in an ongoing 1-year long prospective study. We are closely monitoring muscle and tendon structure at each clinical visit. Once cleared for activity, we will test plantarflexor strength and power in these patients using dynamometry and single-leg heel raises (Baxter & Piazza, 2014; Baxter, Hullfish & Chao, 2018). Prior research also suggests that these structural changes in muscle are permanent (Baxter, Hullfish & Chao, 2018) and the magnitude of these changes may be predictive of long-term functional deficits (Silbernagel, Steele & Manal, 2012; Baxter, Farber & Hast, 2018).

Conclusions

In summary, our findings demonstrate that the medial gastrocnemius muscle undergoes rapid changes in configuration following an acute Achilles tendon rupture, which may explain long-term functional deficits in many patients. While the mechanisms governing these changes remain unclear, small animal studies suggest that perturbations in muscle-tendon tension stimulate these changes. Future research should be focused on the effects of muscle-tendon tension on muscle remodeling pathways.

Supplemental Information

Supplemental Information 1 Muscle architecture data measured at each study time point and statistical output.

Medial gastrocnemius muscle architecture measurements at each study time point with ultrasound imaging provided in this spreadsheet. Raw data and statistical outputs are included for the comparisons between injured limb data at each time point and the contralateral (control) limb at each study time point (“Muscle Architecture” sheet). The reliability of our measurements of the contralateral (control) limb muscle at each visit shows that the coefficient of variation is below 0.1 and intra-class correlations are all very good (> 0.84) for all measurements (“Control Reliability” sheet).

Click here for additional data file.

Additional Information and Declarations

Competing Interests

Author Contributions

Human Ethics

Data Availability

The authors declare that they have no competing interests.

Todd J. Hullfish performed the experiments, analyzed the data, contributed reagents/materials/analysis tools, prepared figures and/or tables, authored or reviewed drafts of the paper, approved the final draft.

Kathryn M. O’Connor conceived and designed the experiments, authored or reviewed drafts of the paper, approved the final draft.

Josh R. Baxter conceived and designed the experiments, analyzed the data, contributed reagents/materials/analysis tools, prepared figures and/or tables, authored or reviewed drafts of the paper, approved the final draft.

The following information was supplied relating to ethical approvals (i.e., approving body and any reference numbers):

The University of Pennsylvania granted Institutional Review Board approval (828374).

The following information was supplied regarding data availability:

The raw data are available in the Supplemental Files.

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
