# Peer review of "Gastrocnemius fascicles are shorter and more pennate throughout the first month following acute Achilles tendon rupture"

_PeerJ, doi:10.7717/peerj.6788_

## Round 0.1 · original submission · Major Revisions

We have received 3 constructive reviews of the MS, which have many tips for revising it. They agree that there is a publishable paper here, with some effort. There is a more major criticism of the MS's claim from Reviewer 2 that must be dealt with carefully, regarding the comparison of opposite vs. injured limb and claim of changes in the injured limb vs. pre-injury. Please address all points individually and include a Tracked Changes MS for me and reviewers to check on resubmission. Thank you!

Reviewer 1 ·

Basic reporting

Thank you for letting me review this interesting and novel paper. The manuscript ”Gastrocnemius fascicles are shorter and more pennate throughout the first month following acute Achilles tendon rupture” is well written and may add some new, interesting facts in this field. The purpose of this study was to quantify the immediate structural changes to the medial gastrocnemius the first 4 weeks after an Achilles tendon rupture.
However, there is a need for some revisions, especially in the Conclusion and Discussion section in order to avoid too much speculation.

Experimental design

The design of this study is adequate and described in a proper way. A power analyze has been performed that showed that 10 subjects were needed for detect a 10% decrease in fascicle length. The coefficient of variation was calculated to be less than 10%. There is a lack in explanation how the subjects in this particular study were recruited from the mentioned 1-year long prospective study. This needs to be clarified.

An advantage in this paper is that the same person performed all the ultrasound measurements. The coefficients of variation between the measurements on the healthy limb have been calculated and they were found to be less than 10%. However, this says nothing about the normal variation between the fascicle length between the healthy and the normal side. There is a need to clarify the facts in this matter.

Validity of the findings

The data in this study is well presented and there is also supplementary material to clarify all the measurements.
However, the conclusion in the abstract and in the Conclusion section are not expressed in the same way. In my opinion, the authors can not draw the conclusion that “careful immobilization paradigms may prove to be effective for preserving muscle structure and quality in patients following tendon injuries” from the results in this fairly small study performed during 4 weeks. It is important for the authors to keep in mind what variables you used for power analysis before drawing any far-reaching conclusions. A suggestion is to use the last sentence in the abstract for Conclusion.

Additional comments

Sometime you use the word “pennation” and sometimes “pennation angle” in the paper. Please, use the same word throughout the manuscript.

Abstract
Line 21-25: Please clarify that the measurements refer to the injured limb and (I suppose) in comparison with the healthy limb.
Introduction
Line 69-80: This paragraph is the purpose and the hypothesis. This section needs to be re-written since there are some difficulties to follow what is the purpose/hypothesis and what is more background. A suggestion is to put line 69-73 at the very end.

Methods
Line 86: You have listed the inclusion criteria. Were there any exclusion criteria? I miss the inclusion criteria in to this study from the “1-year long prospective study” (line 101).

Line 96-99: What are normal differences between limbs (in healthy subjects) in fascicle length, pennation angle, muscle thickness and echogenicity?

Line 109-124 For proper intra-reliability calculations between measurement, there may be a need for ICC-values for the measurements with ultrasound.

Line 114: How did you confirm that you measured in the middle of the muscle belly? Please clarify.

Line 124: The coefficient of variation is often expressed in percent. This is just a suggestion and not necessary to change.

Line 134: How did you define “poor quality”? Please, clarify.

Results
The results are presented in a clear and consistent way that make it easy for the reader to follow.
I know that you present your measurements in percent, but what unit has echogenicity?

Discussion
In this section, I miss a discussion of your results in the aspects of:
- Treatment with or without surgery
- Gender differences (you have only evaluated men)
- How is the fascicle length in the beginning of the injury since the mean time before the first evaluation was 4,5 days? This is linked to the question about how much is the normal difference in fascicles length between limbs in healthy subjects.


Line 163-171: This paragraph is more like a continuing of the Introduction. A suggestion is that you start with your major findings (like you write at line 172-174).

Line 174: How do you know that the changes is a mechanical response to a sudden loss of intra-muscular tension? This sentence is more of a speculation than a conclusion of your results.

Line 179: I agree that your results are an important advancement in your study, but not (at least yet) in the treatment of tendon injuries.

Line 182: Please clarify that you are referring to the injured gastrocnemius.

Line 202: You do not need to refer to your own figure in the Discussion section.

Conclusion
The conclusion in the abstract and in the Conclusion section are not expressed in the same way. In my opinion, the authors can not draw the conclusion that “careful immobilization paradigms may prove to be effective for preserving muscle structure and quality in patients following tendon injuries” from the results in this fairly small study. A suggestion is to use the last sentence in the abstract for Conclusion.

Please, keep in mind what variables you used for power analysis before drawing any far-reaching conclusions.

Figures
Figure 2: If possible, it would be valuable to present a bigger picture. About the figure legend: Here, there is no need to write that the same investigator performed all evaluations. It would be valuable to know where (on the medial gastrocnemius muscle) the imaging is taken.

·

Basic reporting

The language is clear and professional. Literature references and background/context are provided for the relevance of Achilles tendon injuries and plantar flexor function, whereas information is missing to explain why the chosen time period of the first four weeks after injury is relevant.

Experimental design

The design of the experiment is solid but I believe originality and relevance of the study could be greatly improved by reporting the follow-up of the patients over a longer time period. Other variables, for example soleus muscle architecture, would also strengthen the paper since soleus is a primary contributor to ankle joint torque during locomotion (e.g. Hamner 2018 J Biomech). Although research questions are well defined by the authors, they do not convincingly convey the relevance of the work.

Validity of the findings

The conclusion that muscle architecture changed because of the tendon rupture is not directly supported by the data and that should be acknowledged by the authors. To conclude this, it would have been necessary to perform the same measurements before injury. The present study-design allows only to conclude that there are differences between the injured and the healthy leg. Whether these differences resulted from the tendon rupture or whether they existed before and possibly have contributed to a higher injury risk, is unclear from the present data.

Additional comments

The authors explain that no functional tests can be performed within this time frame, so I am wondering why the later follow-up measurements of the patients were not reported to relate changes in muscle architecture to function and to see whether the short term changes persist. The authors could not convince me why changes in muscle architecture are important in the first four weeks after injury when the injured side cannot be loaded. I recommend to add measurements over a longer period and to test functional outcomes as early as possible before resubmitting.

Reviewer 3 ·

Basic reporting

No comment

Experimental design

In this study, the authors determine the effects of Achilles tendon rupture on medial gastrocnemius muscle architecture. They find that an acute AT rupture elicits immediate changes in muscle fascicle length and pennation angle, that persist for up to 4 weeks post injury. The authors hypothesize that these structural changes occur to maintain MTU tension following injury. The manuscript is well-written, however as discussed below, you will see that I have a few concerns related to the experimental protocols and methodologies that I hope can be addressed by the authors.


Major Comments:

Methods – Did you, and if so how, ensure that ankle angle was consistent between the injured and uninjured limbs when acquiring ultrasound images? This is critical to ensuring that structural changes are due to mechanical changes rather than kinematics. Also, did you measure this angle at each testing session and ensure that it was the same at week 0, 2, and 4 for each participant. For these comparisons to be made, ankle angle must be controlled. The authors state ‘with their feet and ankles supported by the edge of the table in PF and kept in the same position for all imaging sessions’. Please report what this plantarflexion angle was and also how you ensured it was in the same position for each participant across testing sessions.

Comparisons to control limb – it is unclear why the authors used the uninjured limb at week 0 as the control for all comparisons. Given that it is likely for de-training effects to have also affected muscle structure on the uninjured limb since participants would not be exercising between weeks 0 and 4, the authors should compare structural changes between the injured and uninjured limb at EACH time point, in an attempt to remove the confounding effects of de-training/exercise. I believe the authors should have these data given that they confirmed probe placement repeatability measuring the control limb at each visit.

Reliability of ultrasound measurements – the authors state that many of the ultrasound images show to be a poor quality and should conduct and report intra or inter-rater reliability measures on these data.

Figures 1 and 2 are poor quality and not informative. I suggest changing these. See methods section.

Validity of the findings

No comment

Additional comments

Minor Comments:

Abstract

Line 14 – ‘on medial gastrocnemius’ should be ‘on medial gastrocnemius structure’

Line 16 – move B-mode from Line 18 to Line 16 when ultrasound is first discussed

Line 19 – was ‘initial clinical treatment’ always the first image acquisition? Or did patients receive clinical treatment before this first time point of data?

Line 27 – I suggest beginning this sentence with ‘Our results suggest…’ or a similar statement and please replace ‘configuration’ structure or architecture to be consistent in your language throughout.

Introduction:
Line 44 – Please add in whether decreased resting ankle angle refers to more PF or DF, as sign conventions are often different across the literature.

Line 45 – what is calf raise ‘function’ referring to? Force, kinematics, number of reps?

Line 49 – Highlighting that these results are from animal models would be useful to the reader

Line 54 – ‘PF function is governed by …. and mechanical properties’. Mechanical properties of what? (Muscle, tendon?)

Line 66 – modeling should be remodeling

Line 68 – have should be has

Line 72 – ‘and would persist’ should be ‘and these changes would persist’


Methods

See above major concerns regarding limb configuration during image acquisition, comparison to control image and reliability of US measurements.

Line 99 – requires a reference.

Methods – states that ultrasound video sequences were acquired but single images were analyzed. How did you choose which image to analyze?

Line 142 – did you also run the t-test of the echogenicity data? This is not included.

Figure 1 is not a very informative figure (and the image quality of the leg traces is quite poor). I suggest replacing Fig 1 with a more powerful methods figure that has either a cartoon or camera image of a person in the position which you acquired images. You can still include the details within your current Figure 1, but this is already within the text more than once (and can remain in the figure caption as is).

Figure 2 – I would like to see the authors choose one representative subject and present the control and the injured ultrasound images together (A and B) with the representative fascicle, pennation and thickness outlined on each of the images. Further, in this figure the line drawn is not on the deep aponeurosis.

Discussion

Line 176 – biologic should be biological

Line 195 – compares should be compare

Line 244 – the authors mention that they are prospectively following these patients for 1 year to link changes in muscle structure with tendon elongation and functional deficits. Why was tendon length during these time 0, 2, and 4 week time points not measured and reported. See reference below for ultrasound – tape measure method.

Barber, L., Barrett, R., & Lichtwark, G. (2011). Validity and reliability of a simple ultrasound approach to measure medial gastrocnemius muscle length. Journal of anatomy, 218(6), 637-642.

From Fig 4 it is apparent that muscle structure, especially pennation angle, changes from week 2 to week 4. Can you authors include some discussion regarding this change. What changes from a cast to a boot may lead to alterations in muscle structure? (Weight bearing, activity level?)

---

## Round 0.2 · Major Revisions

Three reviewers have again checked the MS, and views vary. One remains unconvinced that there is value to the study. Another has major concerns about how the limited patient dataset was selected from a broader future study's sample. The final reviewer has three substantial concerns about ultrasound imaging and data representation. These overall constitute a need for further major revision. However, it is possible that substantial revisions and a clear Response might obviate further review (to avoid more reviewer-fatigue), but this is far from guaranteed. Overall, however, there remains interest in the study and it could fit the journal's criteria for publication if it is deemed scientifically sound after revision.

Reviewer 1 ·

Basic reporting

Thank you for letting me re-review the manuscript “Gastrocnemius fascicles are shorter and more pennate throughout the first month following acute Achilles tendon rupture”.

The manuscript has improved significantly. I have only one major concern; I think it already in the Method section (Study design) should be clarified that the findings in the present study is a preliminary data set from a larger clinical cohort. Now, the reader is informed in the last paragraph in the Discussion section which may be confusing (line 280-284). In my opinion, there is also a need for clarification how these 10 patients were chosen from the larger dataset? Consecutively?

Experimental design

No comment

Validity of the findings

No comment

Additional comments

No comment

·

Basic reporting

I understand the authors position that they don’t have the relevant data available yet and that is regrettable because of the limited information brought by the current data.

Experimental design

no comment

Validity of the findings

no comment

Reviewer 3 ·

Basic reporting

The authors have done a nice job in revising their manuscript in response to the minor comments and suggestions, however, I still have 3 major concerns that should be addressed.


Major Comments:

(1)
Reviewer: Methods – Did you, and if so how, ensure that ankle angle was consistent between the injured and uninjured limbs when acquiring ultrasound images? This is critical to ensuring that structural changes are due to mechanical changes rather than kinematics. Also, did you measure this angle at each testing session and ensure that it was the same at week 0, 2, and 4 for each participant. For these comparisons to be made, ankle angle must be controlled. The authors state ‘with their feet and ankles supported by the edge of the table in PF and kept in the same position for all imaging sessions’. Please report what this plantarflexion angle was and also how you ensured it was in the same position for each participant across testing sessions.

Author: We thank the Reviewer for this comment and agree that ankle angle during imaging is indeed an important consideration. During the start of each image acquisition, patients lay on a treatment table with their feet freely hanging off the edge of the table. Next, the patients were carefully shifted onto the table while preserving the resting ankle angle of the uninjured leg. We have used a similar approach for longitudinally studying Achilles tendon properties with ultrasound (Kagan et al). Due to time constraints, we decided against measuring this ankle angle but did confirm that muscle structure measurements had low coefficients of variation and intra class correlations between the three scanning sessions for the uninjured side (< 10%). This has been described in our methods section (Lines 97-102, 128-129).

Reviewer: This needs to be added to limitations – as visual inspection is not the standard in literature measuring plantarflexor resting muscle architecture. Further, while this may allow for ‘preserving’ ankle angle during one testing session, it does not allow for a constant angle ankle across testing sessions, which is of critical importance for the comparisons being made in this study. I urge the authors to carefully control this (or measure using a goniometer) in future experiments.

(2)
Reviewer: Comparisons to control limb – it is unclear why the authors used the uninjured limb at week 0 as the control for all comparisons. Given that it is likely for de-training effects to have also affected muscle structure on the uninjured limb since participants would not be exercising between weeks 0 and 4, the authors should compare structural changes between the injured and uninjured limb at EACH time point, in an attempt to remove the confounding effects of de-training/exercise. I believe the authors should have these data given that they confirmed probe placement repeatability measuring the control limb at each visit.

Author: We appreciate this comment and would like to clarify our decision to compare the injured scans to the week 0 injured scans. First, our hypothesis was that Achilles tendon ruptures result in differences in muscle structure from baseline data. While it is possible that the patients underwent some de-training effects, our high ICC and low coefficients of variation in the control muscle measured at each clinical visit suggest that de-training did not occur. We have provided additional metrics of measurement variability between clinical visits for the control muscle to highlight the consistency in this measurement (Lines 128-130).


Reviewer: I would like to see a supplemental figure of Figure 3 where the measures are compared to the control limb at each time point --given that the authors have collected this data. Although the coefficient of variation is low (less than 10%), the change in fascicle length from control to week 0 is 15 %.


(3)
Reviewer: Reliability of ultrasound measurements – the authors state that many of the ultrasound images show to be a poor quality and should conduct and report intra or inter-rater reliability measures on these data.

Author: We apologize for the confusion created by this previous wording. We noticed that muscle quality on the injured side appeared to be ‘poor quality’. This was evident by a lack of contrast between the fascicles and interstitial space. To preserve consistency between scan sessions and subjects, we kept all scan parameters constant. A previous report by Fukumoto et al (2012) that found echo intensity correlates with muscle strength and suggested that this was caused by increased fibrous and adipose tissues within the muscle belly. This agreed with our initial intuition of our observations so we decided to include this ultrasound analysis as a means of quantifying ‘muscle quality’. We have clarified this in the methods section (Lines 141-142).

Reviewer: I can see from figure 2 that the image quality, as evident by the contrast between fascicles and connective tissues to be very poor. Ensuring scan parameters were consistent is useful for the echo intensity analysis but not for the fascicle architecture analysis, as presumably you could have optimized these parameters for each participant (ie: changed power, gain, focus depth, etc). My concern is that the quality of the images make it challenging to track fascicles, and I would like to see an inter-rater reliability testing conducted to ensure that fascicle tracking is consistent between raters.

Experimental design

no comment

Validity of the findings

no comment

---

## Round 0.3 · Minor Revisions

The reviewers are much more satisfied with the MS and there are just 2 easy changes to make, then the MS should be acceptable. Thanks for your patience!

Reviewer 1 ·

Basic reporting

No comment

Experimental design

No comment

Validity of the findings

No comment

Additional comments

Thank you for letting me re-re-review your manuscript “Gastrocnemius fascicles are shorter and more pennate throughout the first month following acute Achilles tendon rupture”.
Thank you also for your clarifying answer to my concern about the inclusion criteria in this study. I no longer have any comments about your study.

Reviewer 3 ·

Basic reporting

I have two remaining comment that I would like to see addressed.

(1) As the authors acknowledge, the lack of controlling for ankle angle is a valid concern. However, this has not been included within the limitations section of the manuscript as suggested in the previous review. Please include these details, along with the potential effects on results and interpretation of results, within your discussion

(2) The new figure 3 caption is incorrectly matched with the figure labels. (A) is echogenicity in the figure but in the caption is refers to (A) as fascicle length. It appears that A and D need to be swapped.

Experimental design

NA

Validity of the findings

NA

Additional comments

NA

---

## Round 0.4 · accepted · Accept

Well done! No further changes needed.